# A single dose of inactivated influenza virus vaccine expressing COBRA hemagglutinin elicits broadly-reactive and long-lasting protection

Hua Shi[1], Xiaojian Zhang[1], Ted M. Ross[1,2,3,4]*

1 Center for Vaccines and Immunology, University of Georgia, Athens, GA, United States of America,
2 Florida Research and Innovation Center, Cleveland Clinic, Port Saint Lucie, FL, United States of America,
3 Department of Infectious Diseases, University of Georgia, Athens, GA, United States of America,
4 Department of Infection Biology, Lehner Research Institute, Cleveland Clinic, Cleveland, OH, United States of America

* rosst7@ccf.org

**Data Availability Statement:** All data are in the manuscript and/or supporting information files.

**Funding:** This project has been funded as part of the Collaborative Influenza Vaccine Innovations

## Abstract

Influenza virus infections present a pervasive global health concern resulting in millions of hospitalizations and thousands of fatalities annually. To address the influenza antigenic variation, the computationally optimized broadly reactive antigen (COBRA) methodology was used to design influenza hemagglutinin (HA) or neuraminidase (NA) for universal influenza vaccine candidates. In this study, whole inactivated virus (WIV) or split inactivated virus (SIV) vaccine formulations expressing either the H1 COBRA HA or H3 COBRA HA were formulated with or without an adjuvant and tested in ferrets with pre-existing anti-influenza immunity. A single dose of the COBRA-WIV vaccine elicited a robust and broadly reactive antibody response against H1N1 and H3N2 influenza viruses. In contrast, the COBRA-SIV elicited antibodies that recognized fewer viruses, but with R-DOATP, its specificity was expanded. Vaccinated ferrets were protected against morbidity and mortality following challenge with A/California/07/2009 at 14 weeks post-vaccination with reduced viral shedding post-infection compared to the naïve ferrets. However, the COBRA-IIVs did not block the viral transmission to naïve ferrets. The contact infection induced less severe disease and delayed viral shedding than direct infection. Overall, the COBRA HA WIV or the COBRA HA SIV plus R-DOTAP elicited broadly reactive antibodies with long-term protection against viral challenge and reduced viral transmission following a single dose of vaccine in ferrets pre-immune to historical H1N1 and H3N2 influenza viruses.

**IMPORTANCE** The goal of the next-generation influenza vaccine is to provide broadly reactive protection against various drifted influenza strains. With the previous studies evaluating the COBRA HA-based vaccines, the breadth of antibody activities was confirmed following two or three vaccinations. However, for the commercial influenza vaccine, only one shot is required. In this study, only one shot was administrated to the pre-immune ferrets and the COBRA-WIV efficiently elicited broadly reactive antibodies and long-lasting protection against the pdm09 strain. Moreover, this study showed that different infection methods can lead to different disease severity, which emphasizes the significance of the model

Centers (CIVICs) by the National Institute of Allergy and Infectious Diseases, a component of the NIH, Department of Health and Human Services, under contract 75N93019C00052. TMR is also supported in part as a Georgia Eminent Scholar by the Georgia Research Alliance, GRA-001. The funders had no role in study design, data collection and analysis, decision to publish, or preparation of the manuscript.

**Competing interests:** The authors have declared that no competing interests exist.

selection. The infection was conducted 14 weeks post-vaccination to evaluate the long-term protection elicited by only one vaccination. This is the first longevity study describing the immune responses elicited by COBRA-IIVs in ferrets and provides promising results for the potential clinical utilization.

## Introduction

Influenza viruses are a worldwide public health threat that causes respiratory tract infections and, in more severe cases, pneumonia and death [1]. Each year, influenza virus infections are responsible for over 5 million hospitalizations for adults and 10.1 million influenza virus-associated acute lower respiratory infections with 34,800 reported deaths for children under 5 years old globally [2, 3]. The Centers for Disease Control and Prevention (CDC) recommends prophylactic vaccination as the most effective method for preventing influenza virus breakouts (https://www.cdc.gov/flu/season/faq-flu-season-2023-2024.htm). In recent flu seasons, the most commonly used seasonal influenza vaccine is the split inactivated vaccine (SIV) consisting of one H1N1 and one H3N2 influenza A virus strain, and two influenza B virus strains selected before the flu season [4]. However, the vaccine effectiveness (VE) was 40%-50% for the recent vaccines [5, 6], and could be much lower (only 19% for the 2014–2015 vaccine) when a mismatch between the vaccine strains and circulating strains occurs [7]. Developing a universal influenza vaccine (UIV) that can provide broad protection against multiple strains for all populations is an unmet goal for the next-generation influenza vaccines.

Hemagglutinin (HA) binds to the sialic acid and facilitates the entry of virions [8], serving as the primary target for the vaccine design [9–11]. To design the HA structure that can elicit broadly reactive antibodies against various influenza strains, the computationally optimized broadly reactive antigen (COBRA) technology was developed by using multiple layered consensus building [12]. COBRA H1, H2, and H5 have been designed in recombinant HA (rHA) or virus-like-particle (VPL) formats, and their effectiveness in eliciting antibodies with a broad spectrum has been confirmed in mouse and ferret models [12–15].

In this study, the inactivated influenza vaccine (IIV) expressing COBRA HA was assessed as a UIV candidate. The whole inactivated influenza vaccine (WIV) was first used in humans to prevent influenza viral infection in 1940 [16], but later the SIV replaced the WIV in the U.S. market for its improved safety [17, 18]. In a previous study, the bivalent COBRA-SIV carrying both COBRA H1 and H3 was designed and tested in naïve ferrets. The results showed that the COBRA-SIV elicited broader or similar antibody reactivities compared to the wild-type (WT) SIV [19]. However, the COBRA-SIV failed to elicit protective antibody levels against all strains in the panels, which could be because of the reduced immunogenicity of SIV. Later, the WIV or SIV expressing an H1 (Y2) or H3 (J4) COBRA HA antigen was designed and tested in mice models for a more comprehensive evaluation of the COBRA HA-based IIVs [20]. The results showed that the addition of an adjuvant enhanced the long-term protection elicited by both COBRA-WIV and COBRA-SIV and the pre-immunity further expanded the antibody activity spectrum [20].

To further evaluate the effectiveness of COBRA-IIVs in eliciting antibodies with broad spectrum under different formulations and restricting viral transmission, herein, the ferret transmission study is designed. The adjuvants are commonly used in vaccines to improve the VE, for example, the adjuvanted influenza vaccine was approved for the elderly in the U.S. [21]. Therefore, in this study, two adjuvants, AddaVax and R-DOTAP, served as active

competitors for optimizing the design of COBRA-IIVs. AddaVax, as an equivalent of the commercial influenza vaccine adjuvant MF59, can stimulate a balanced Th1/Th2 activation [22]. The novel cationic nanoparticle R-DOTAP can stimulate cellular immunity, especially CD8T cells, to eliminate intracellular pathogens [23, 24]. In the previous study, the primary vaccination of COBRA-IIVs successfully recalled memory B cells in the pre-immune mice [20], which indicates that a single dose of COBRA-IIV is sufficient to elicit immune memory. To better evaluate the effectiveness of a single dose of a COBRA-IIV vaccine, pre-immune ferrets were vaccinated with either COBRA-WIV or COBRA-SIV vaccine followed by an H1N1 influenza virus challenge 14 weeks later. Ferrets were assessed for high titer antibodies with HAI activity, protection against direct infection, and prevention of viral shedding and transmission.

## Materials and methods

### Vaccine preparation

The recombinant viruses expressing COBRA HA Y2 (H1N1) or J4 (H3N2) were rescued by collaborators in St. Jude Children's Hospital by using the eight-plasmid reverse transgene system and then inactivated and split by us to generate the COBRA-WIV and COBRA-SIV. This method has been described previously. Briefly, the mixture of 293T (6.25x 10^5 cells/well) and Madin-Darby canine kidney (MDCK) cells (3.2510^5 cells/well) was seeded in 6-well cell culture plates until they achieved 80–90% confluency. The cDNA of each segment of A/Puerto Rico/8/1934 (PR8) was inserted into the plasmid vector pHW2000, except for the segment encoding the HA. It was substituted with the gene encoding the COBRA HAs. Plasmids for virus recovery were prepared meticulously, ensuring each attained a concentration as close to 1μg/ml as feasible. 1μg of each plasmid was added to each well (8μg in total) mixed with 500μL OptiMEM (ThermoFisher, 31985–062) and 16μL TransIT-LT (Mirus, MIR2300), followed by overnight incubation at 37°C with 5% CO2. The subsequent day involved replacing the medium with OptiMEM containing TPCK-Trypsin (Worthington, LS003740). As the days progressed, diligent monitoring for cytopathic effects (CPE) and hemagglutination (HA) was necessary. If required, blind passages were conducted, and the supernatant was stored at -80°C for future use.

To amplify the rescued virus, MDCK cells were prepared and when they achieved 80–90% confluency in the T-175 cell culture flask (ThermoFisher, 159910), the cell culture media was discarded and replaced with the viruses (MOI: 0.01) diluted in 1XMEM (Corning). For each virus, approximately 20 flasks were used to generate the sufficient amount of viruses. After 48–72 hours of incubation at 37°C with 5% CO2, the cells were monitored for CPE, and when obvious CPE was observed, the cell culture media was collected and centrifuged at 2000xg and 4°C for 10 minutes to separate the supernatant. The supernatant was carefully collected and mixed with a 0.5M disodium phosphate (DSP) solution in a ratio of 1 part DSP solution to 38 parts supernatant. Next, 2% BPL (beta-propiolactone) was added dropwise to the DSP-supernatant mixture in a ratio of 1 part BPL to 38 parts DSP-supernatant solution, resulting in a final BPL concentration of 0.05% (v/v). The mixture was then incubated on ice with intermittent shaking for 30 minutes before being transferred to a 37°C water bath for 2 hours with periodic mixing. Following incubation, the pH of the mixture was adjusted to 7.3–7.4 by using a 7% sodium bicarbonate solution, monitored with pH strips. The resulting antigen are the COBRA-WIV. They were quantified by anti-H1 or anti-H3 western blot and stored at -80°C until further use. Additionally, an inoculation test was carried out to detect the presence of any live viruses, providing further assurance of the antigen's safety for use in subsequent applications.

To produce the COBRA-SIV, a 1% (v/v) final concentration of Triton X100 was introduced into the WIV solution, followed by incubation at room temperature for one hour. Subsequently, to ensure safety, 0.1g of BioBeads (Bio-Rad) per mL of the mixture was incorporated to eliminate excess Triton X100. The mixture underwent gentle shaking on a shaker at 4˚C overnight, after which the liquid component was separated and collected. In this way, the COBRA-SIV was generated. A similar western blot was conducted to quantify the HA content and then the COBRA-SIV was labeled and stored at -80˚C for further use.

## Animal, pre-infection, and vaccination

Female Fitch ferrets (Mustela putorius furo) aged 6 to 15 months were obtained from Triple F Farms (Gillett, PA, USA) after de-scenting and spaying procedures. They were resting in the animal facility for one week before any procedures. Blood samples were collected as the baseline for each ferret's serological status regarding the A/California/07/2009 H1N1 influenza virus. Ferrets were housed in pairs with access to food, water, and enrichment when not undergoing procedures. Anesthesia using vaporized isoflurane was administered before any interventions such as bleeds, vaccination, infection, nasal washes, or euthanasia. Ethical standards, as outlined in the Guide for the Care and Use of Laboratory Animals, Animal Welfare Act, and Biosafety in Microbiological and Biomedical Laboratories (AUP: A2020 11-016-Y1-A6), were strictly followed during all animal procedures. To establish pre-immunity, 42 naïve ferrets were intranasally infected with the Sing/86 and Pan/99 influenza virus ($5*10^5$PFU/virus) eight weeks prior to the vaccination. Following this, ferrets were vaccinated intramuscularly in the thigh muscle with vaccines (7.5μg/antigen) formulated as described in Table 1 or PBS serving as the mock-vaccinated group. The Addavax adjuvant (InvivoGen, San Diego, CA, USA) or R-DOTAP (PDS, Florham Park, NJ, USA) was mixed in a 1:1 ratio with sterile PBS and antigens. Mock-vaccinated groups received 500μL of sterile PBS. Blood samples were collected in BD Vacutainer SST tubes 2-week post-preinfection to confirm the pre-existing antibodies specific to Sing/86 and Pan/99 (the seroconversion is shown in S1 Fig); and 2-, 10-, and 14-week post-vaccination. Serum separation was achieved by centrifugation at 2500 rpm for 10 minutes following 30 minutes of incubation at room temperature. The purified serum was then stored at −20˚C until further analysis. Besides, the other 18 ferrets remained naïve and unvaccinated for the whole time serving as naïve transmitters (n = 9) or naïve receivers (n = 9) in the transmission study. The experimental timeline is shown in Fig 1.

All animals were humanely euthanized with B-euthanasia under anesthesia either at the end of the study or whenever a cumulative clinical score of three was reached. Clinical signs with their scores were as follows: nasal discharge/sneezing/diarrhea (0.5; not used for humane endpoint calculation but used for graphical representation), lethargy (1), dyspnea (2), cyanosis (2), neurological signs (3), moribund (3), laterally recumbent (3), failure to respond to stimuli (3), weight loss of 20–25% (2), and weight loss of greater than 25% (3). The maximum of the two clinical scores recorded for each day was used for analysis.

**Table 1. Formulation information for the COBRA-IIVs.**

| Group | Vaccine | Dose | Adjuvant |
|---|---|---|---|
| 1 | COBRA Y2+J4 WIV | 7.5ug/HA/ferret | N/A |
| 2 | COBRA Y2+J4 WIV | 7.5ug/HA/ferret | AddaVax |
| 3 | COBRA Y2+J4 WIV | 7.5ug/HA/ferret | R-DOTAP |
| 4 | COBRA Y2+J4 SIV | 7.5ug/HA/ferret | N/A |
| 5 | COBRA Y2+J4 SIV | 7.5ug/HA/ferret | AddaVax |
| 6 | COBRA Y2+J4 SIV | 7.5ug/HA/ferret | R-DOTAP |

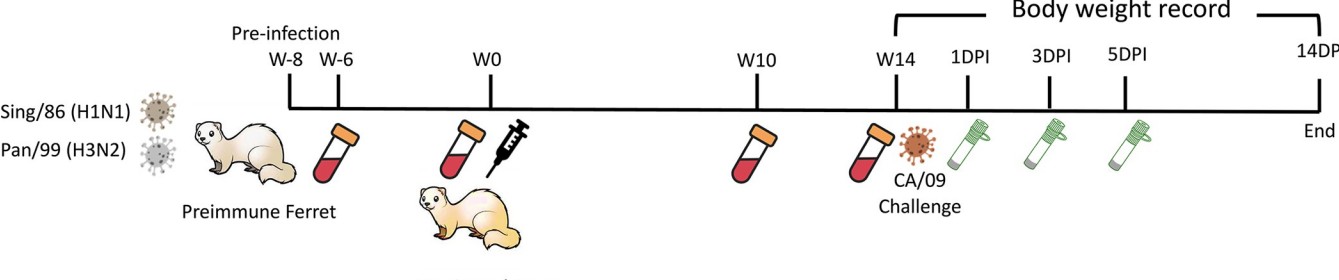

**Fig 1. Experimental timeline for vaccination and direct challenge.** 42 naïve ferrets were pre-infected with Sing/86 and Pan99 ($5*10^5$PFU/virus/ferret) 8 weeks prior to the vaccination. Sera samples were collected 2 weeks post-pre-infection. At week 0, ferrets were evenly divided into 6 vaccinated groups (n = 6/ group) receiving vaccines named: COBRA-WIV, COBRA-WIV plus AddaVax, COBRA-WIV plus R-DOTAP, COBRA-SIV, COBRA-SIV plus AddaVax, and COBRA-SIV plus R-DOTAP; and mock-vaccinated group (n = 6) receiving PBS only. The detailed information of each vaccine's formulation is shown in Table 1. Sera samples were collected on week 2, 10, and 14 to measure the antibody levels. Ferrets were challenged with CA/09 ($1*10^6$PFU/ferret) at week 14, with 6 naïve mock-vaccinated ferrets serving as the mock control. Nasal washes were collected at 1DPI, 3DPI, and 5DPI to monitor the viral shedding. Body weight and clinical signs were closely recorded daily until 14DPI. DPI: day post-infection.

## Viruses

All viral strains employed in this study were sourced from Influenza Reagents Resource (IRR), BEI Resources, the Centers for Disease Control (CDC), or graciously provided by Virapur (San Diego, CA). Those viruses were amplified strictly following the environment of their original stocks. The H1N1 influenza viruses included A/Singapore/6/1986 (Sing/86), A/California/ 07/2009 (CA/09), A/Brisbane/02/2018 (BS/18), A/Guangdong-Maonan/SWL1536/2019 (GD/ 19), and A/Victoria/2570/2019 (Vic/19). The H3N2 influenza virus strains included A/Panama/2007/1999 (Pan/99), A/Switzerland/9715293/2013-mouse adapted (SW/13), A/Hong Kong/4801/2014 (HK/14), A/Singapore/IFNIMH-16-0019/2016 (Sing/16), A/Kansas/14/2017 (Kan/17), A/South Australia/34/2019 (SA/19), A/Hong Kong/2671/2019 (HK/19), and A/Tasmania/503/2020 (TAS/20).

## Direct infection

Ferrets were challenged by CA/09 ($1*10^6$PFU/ferret/1mL) via intranasal inoculation at week 14. Following infection, animals were closely monitored, with observations conducted twice daily to detect any potential clinical signs. Additionally, their weight was measured once daily until 14 days post-infection (DPI). Nasal washes were conducted on days 1, 3, and 5 DPI utilizing 3 mL of sterile PBS each time.

## Contact infection

The contact infection was designed in a two-way transmission: from the naïve transmitter (NT) to the vaccinated receiver (VR) and from the vaccinated transmitter (VT) to the naïve receiver (NR). NT ferrets were infected by CA/09 ($1*10^6$PFU/ferret/1mL) via intranasal inoculation. One day after the infection, the VR ferrets were paired with the NT ferrets. For the other direction of transmission, VT ferrets were infected by CA/09 ($1*10^6$PFU/ferret/1mL) via intranasal inoculation and on 1 DPI, NR ferrets were co-housed with the VT ferrets. The transmission from the NT to the NR serves as the mock control. The NT ferrets were infected by CA/09 ($1*10^6$PFU/ferret/1mL) via intranasal inoculation. One day after the infection, the NR ferrets were paired with the NT ferrets.

All co-housed ferrets stayed together all the time for the rest of the experiment. Following infection, animals were closely monitored, with observations conducted twice daily to detect

any potential clinical signs. Additionally, their weight was measured once daily until 14 days post-infection (DPI). Transmitters' nasal washes were conducted on 1, 3, and 5 DPI, and the receivers' nasal washes were collected on 2 and 4 DPI, utilizing 3 mL of sterile PBS each time.

## Hemagglutination inhibition (HAI) assay

The hemagglutination inhibition (HAI) assay is capable of identifying antibodies that specifically bind to the hemagglutinin (HA) head, thereby preventing the attachment of viral HA to sialic acid receptors on red blood cells. This inhibition results in the prevention of hemagglutination, which can be readily observed. To ensure the assay's specificity, all serum samples underwent treatment with a receptor-destroying enzyme (RDE) (Denka Seiken, Co., Japan). Serum samples were mixed with RDE at a 1:3 ratio and incubated at 37°C overnight. Subsequently, the mixture was subjected to heat inactivation at 56°C for 45 minutes, followed by the addition of phosphate-buffered saline (PBS) to achieve a final ratio of serum: RDE: PBS of 1:3:6. The RDE-treated serum samples were then subjected to serial dilution in PBS within a 96-well v-bottom plate. For H1N1 viruses, turkey red blood cells (TRBC) sourced from Lampire Biologicals, Pipersville, PA, USA, were diluted in PBS to a final concentration of 0.8%. Each virus was adjusted to a concentration of 8 hemagglutination units (HAU)/50μL, and an equal volume of virus was added to the serum plate. Following a 20-minute incubation period, 0.8% TRBC was added to each well, and the plates were gently tapped to ensure thorough mixing before being incubated at room temperature (RT) for 30 minutes. Conversely, for H3N2 viruses, guinea pig red blood cells (GPRBC) from Lampire Biologicals, Pipersville, PA, were diluted in PBS to a concentration of 0.75%. Similarly, each virus was adjusted to a concentration of 8HAU/50μL in the presence of 20nM Oseltamivir, and an equal volume of the virus was added to the serum plate. After a 30-minute incubation period, 0.75% GPRBC was added to each well, followed by gentle mixing and incubation at RT for 1 hour. HAI titers were determined based on the reciprocal dilution of the last well where no hemagglutination was observed. According to guidelines provided by the World Health Organization (WHO), HAI titers exceeding 1:40 are considered indicative of a protective level (Use, 2016).

## Microneutralization assay

The H1N1virus strains CA/09 and Vic/19 and H3N2 strains Sing/16. KS/17, HK/19, SA/19, and TAS/20, with a final concentration of a 100X tissue infectious dosage 50 (TCID50) per 50μL, underwent incubation with diluted mice serum samples in two-fold serial dilution, ranging from 1:10 to 1:1280, within the virus diluent. One hour of incubation at 37°C, 5% CO2 was conducted to allow the neutralization between the antibodies and viruses. Subsequently, the MDCK cells with an adjusted concentration of about $3*10^5$ cells/mL were added to the mixtures of the virus and serum and allowed to an overnight incubation at 37°C, 5% CO2 for viral growth. Following this incubation period, the cell medium was discarded and fixed with 80% Acetone, followed by 3X wash with wash buffer. 100 μl of the primary antibody Rabbit-anti-influenza-A-NP Polyclonal Antibody (Thermo Fisher, PA5-81661) was diluted 1:2000 and added to the fixed cells followed by a one-hour incubation at room temperature. After 3X wash with wash butter, 100 μl of the secondary antibody Goat-anti-rabbit IgG (H+L) HRP (Thermo Fisher, 31460) 1:2000 diluted was added and subjected to a one-hour incubation at room temperature. Upon completion of the incubation period, the plates were washed 5 times, and 100 μl of o-phenylenediamine dihydrochloride (OPD) (Sigma, cat. no P8287) was added, initiating an enzyme reaction. This reaction was stopped after 3 minutes with the addition of 100 μl of 2N sulfuric acid. Quantification of the reaction was performed by measuring the optical density (OD) at a wavelength of 492 nm. The neutralization (NT) titers were determined as

the reciprocal of the serum dilution yielding a reduction in infected cells of $\geq$ 50% compared to the virus control.

## Enzyme-linked immunosorbent assay (ELISA)

ELISA assays were performed to detect the level of IgG against various antigens, following established protocols (Huang et al., 2021). Immulon 4HBX plates (Thermo Fisher Scientific, Waltham, MA, USA) were coated with a solution containing Y2 rHA, J4 rHA, cH6/1, or cH7/3 at a concentration of 1μg/mL. After overnight incubation in a humidified chamber at 4˚C, plates were blocked with 200μL of blocking buffer per well, and incubated at 37˚C for 90 minutes. Concurrently, serum samples were diluted in blocking buffer using a 3-fold serial dilution starting from 1:500. Following blocking buffer removal, 100μL of diluted serum was added to each well. After a 1-hour incubation at 37˚C, a secondary antibody, biotinylated goat anti-ferret IgG (Sigma-Aldrich, St. Louis, MO), diluted in 1:4000, was added to each well (100μL/well) and incubated at 37˚C for 1 hour. Subsequently, 50 μL of ABTS substrate (VWR Corporation) was added to each well and incubated at 37˚C for 15 minutes. The reaction was stopped by adding 50 μL of 1% SDS to each well. Optical density (OD 414) values were measured using a spectrophotometer (PowerWave XS, BioTek) at a wavelength of 414 nm.

## Plaque assay

A plaque assay was conducted according to previously established protocols [13, 14]. Briefly, MDCK cells were seeded at a density of $5 \times 10^5$ cells/well in 6-well plates in cell growth media two days prior to the assay initiation. The assay commenced when cell confluence exceeded 90%. Nash washes were subjected to 10-fold serial dilution ranging from $10^0$ to $10^{-5}$ in Dulbecco's modified Eagle medium (DMEM). These diluted samples were sequentially added to individual wells of 6-well plates and incubated at 37˚C for 1 hour with gentle rocking every 15 minutes. Following virus attachment, the media were aspirated, and the plates were washed twice with DMEM to remove unbound viruses. Subsequently, 2 mL of 0.8% agarose (Cambrex, East Rutherford, NJ, USA) was added to each well, and the plates were further incubated at 37˚C with 5% $CO_2$ for 72 hours. After incubation, the agarose overlay was carefully removed, and the cell monolayers were fixed using a 10% formalin solution, followed by staining with 1% crystal violet (Thermo Fisher). The plates were then rinsed with tap water and allowed to air dry. The number of plaques on each plate was counted, and the plaque-forming units (PFU) per milliliter for each virus were calculated.

## Ethics statement

Animals were cared for under the University of Georgia (UGA) Research Animal Resources guidelines for laboratory animals. All procedures were reviewed and approved by the Institutional Animal Care and Use Committee (IACUC). The study was conducted in accordance with the local legislation and institutional requirements.

## Statistical analysis

Differences in weight loss, dynamic antibody titers against challenge strains, and lung viral titers were analyzed by two-way ANOVA. The differences in HAI titer and IgG titers were analyzed by one-way ANOVA. Statistical significance was defined as a p-value of 0.05. Graphs and statistical analyses were done using GraphPad Prism software.

## Results

### COBRA-IIVs elicited broadly reactive antibodies against multiple H1N1 and H3N2 strains

This study assessed the effectiveness of the COBRA-IIV vaccines in eliciting antibodies with broadly reactive HAI activity following a single vaccination in ferrets with pre-existing immunity to historical H1N1 and H3N2 influenza strains (Fig 1). Eight weeks following infection with historical influenza strains, ferrets were vaccinated with either WIV or SIV vaccine expressing COBRA HA antigens formulated with R-DOTAP or AddaVax adjuvant or nonadjuvanted (Table 1). All pre-infected ferrets showed existing antibodies against Pan/99 and Sing/86 2 weeks after the pre-infection (S1 Fig). Two weeks following vaccination, ferrets had antibodies with HAI activity (Fig 2). The mock vaccinated ferrets had no detectable HAI activity, therefore, only HAI activity is shown for pre-immune, vaccinated ferrets. Ferrets with pre-existing anti-influenza virus immunity had antibodies with HAI activity (average titer ≥1:40) at 2 weeks following COBRA-WIV vaccination against 50% of the H1N1 influenza virus strains and 100% of the H3N2 influenza virus strains (Fig 2A). In contrast, ferrets vaccinated

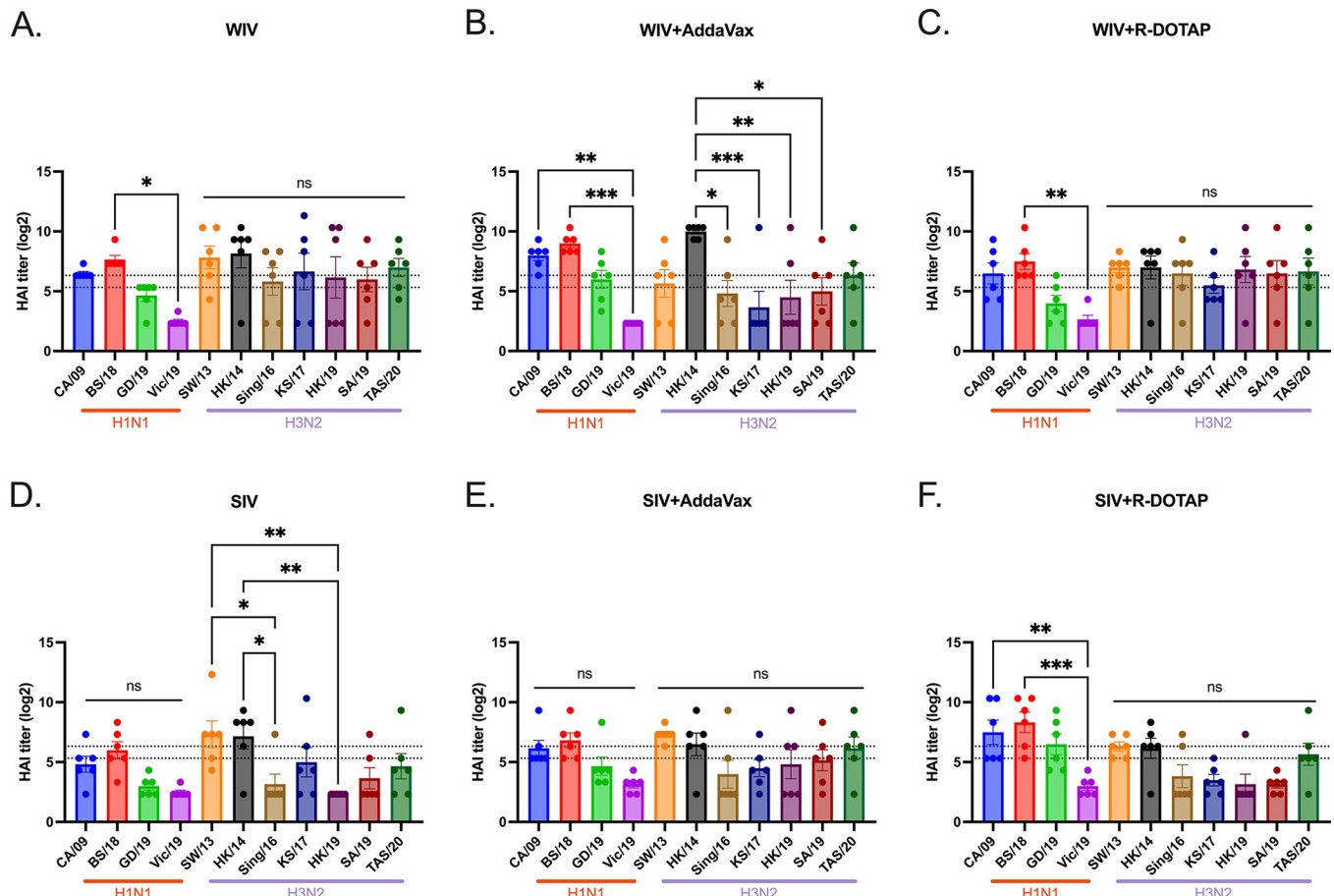

**Fig 2. HAI titers against H1N1 and H3N2 panels for sera collected 2 weeks post vaccination.** The HAI titers elicited by COBRA-WIV (A), COBRA-WIV plus AddaVax (B), COBRA-WIV plus R-DOTAP (C), COBRA-SIV (D), COBRA-SIV plus AddaVax (E), and COBRA-SIV plus R-DOTAP (F) were measured. The X-axis represents different virus strains. The red bar indicates the H1N1 strains and the purple bar indicates the H3N2 strains. The Y-axis represents the Log2 HAI titers with absolute mean values ± SEM. The lower dotted line indicates the HAI titer of 1:40, and the upper dotted line indicates 1:80. A P value of less than 0.05 was defined as statistically significant (*, $P < 0.05$; **, $P < 0.01$; ***, $P < 0.001$; ****, $P < 0.0001$).

with these same vaccines plus R-DOTAP had similar results (Fig 2B), whereas ferrets vaccinated with these same vaccines plus AddaVax adjuvant had antibodies with HAI activity against 75% of the H1N1 influenza virus strains, but only 43% of the H3N2 influenza virus strains (Fig 2C). Ferrets vaccinated with COBRA-SIV alone or with adjuvant had antibodies with HAI activity that recognized fewer H1N1 or H3N2 influenza viruses in the panel compared to their COBRA-WIV counterparts (Fig 2D–2F). HAI antibody titers against CA/09, the influenza challenge strain, were assessed for 14 weeks post-vaccination (S2 Fig). Regardless of the vaccine administered, ferrets had a four-fold decline in HAI activity against CA/09-specific at week 14 compared to week 2 post-vaccination.

## COBRA-IIV vaccines elicited neutralizing antibodies against H1N1 and H3N2 strains

To assess the ability of the vaccine-elicited sera to neutralize virus infection, a microneutralization assay was performed (Fig 3). With one vaccination, COBRA-WIV with or without adjuvant efficiently elicited neutralizing antibodies against CA/09 and Sing/16, whereas COBRA-SIV failed to stimulate neutralization against CA/09 by itself (Fig 3). There were abundant neutralizing antibodies against HK/19, SA/19 and TAS/20 detected post-vaccination, however, the antibody levels showed no significant difference from the mock controls. The high neutralizing antibodies against recent H3N2 strains could be resulted from the pre-immunity. To monitor the long-lasting antibodies, the serum collected 14 weeks post-vaccination was tested against the challenge strain, CA/09 (S1 Table). The results showed that except for COBRA-SIV, other COBRA vaccines all stimulated long-lasting CA/09-specific neutralizing antibodies with only one vaccination. Moreover, by cross comparing the HAI titers and

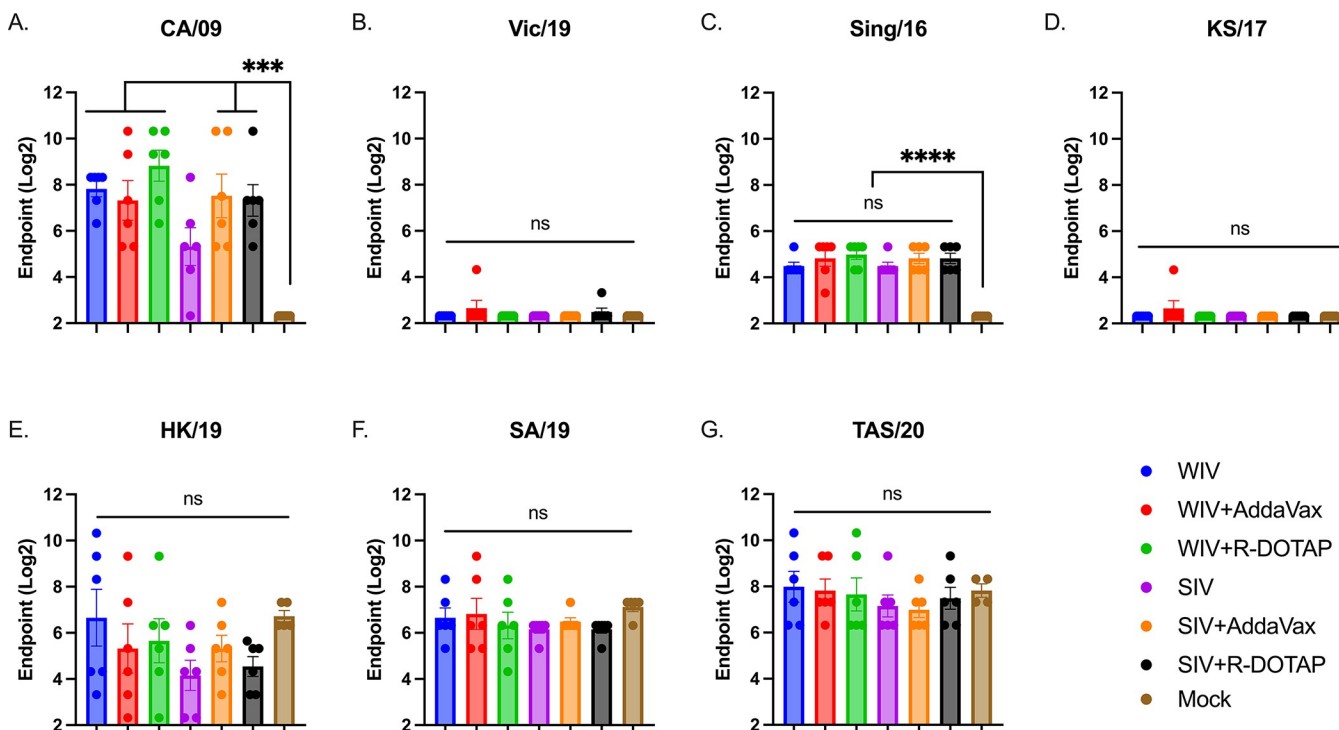

**Fig 3. The breadth of neutralizing antibodies elicited by COBRA-IIVs 2 weeks post vaccination.** The MNA titers elicited by CORBA-IIVs were measured against H1N1 strains CA/09 (A) and Vic/19 (B), and H3N2 strains Sing/16 (C), KS/17 (D), HK/19 (E), SA/19 (F), and TAS/20 (G). The X-axis represents different vaccines. The Y-axis represents the Log2 MNA endpoint with absolute mean values ± SEM. MNA: microneutralization assay.

neutralizing antibody titers in each individual ferrets, it showed that the high HAI titers are associated with high neutralizing antibody titers (S1 Table). However, some samples that showed undetectable HAI activity against CA/09, showed high neutralizing ability for CA/09, which indicates that COBRA-IIVs elicited antibodies targeting epitopes other than the receptor binding sites on HA head domain.

## CORBA-IIV vaccines stimulate the IgG isotype switch and HA stem-binding antibodies

Anti-HA IgG antibody titers were measured 2 weeks post-vaccination (Fig 4). On average, all ferrets vaccinated with the COBRA-WIV vaccine with or without adjuvant had statistically similar anti-H1 HA antibody titers (Fig 4A). Ferrets vaccinated with COBRA-SIV vaccines plus R-DOTAP doubled anti-HA antibody titers compared to ferrets vaccinated with COBRA-SIV vaccine only. All ferrets had statistically similar anti-H3 HA antibodies following vaccination, regardless of the vaccine or adjuvant administered (Fig 4B). To assess the HA stem-binding antibodies, chimeric HA recombinant proteins (cH6/1: chimeric HA with H6

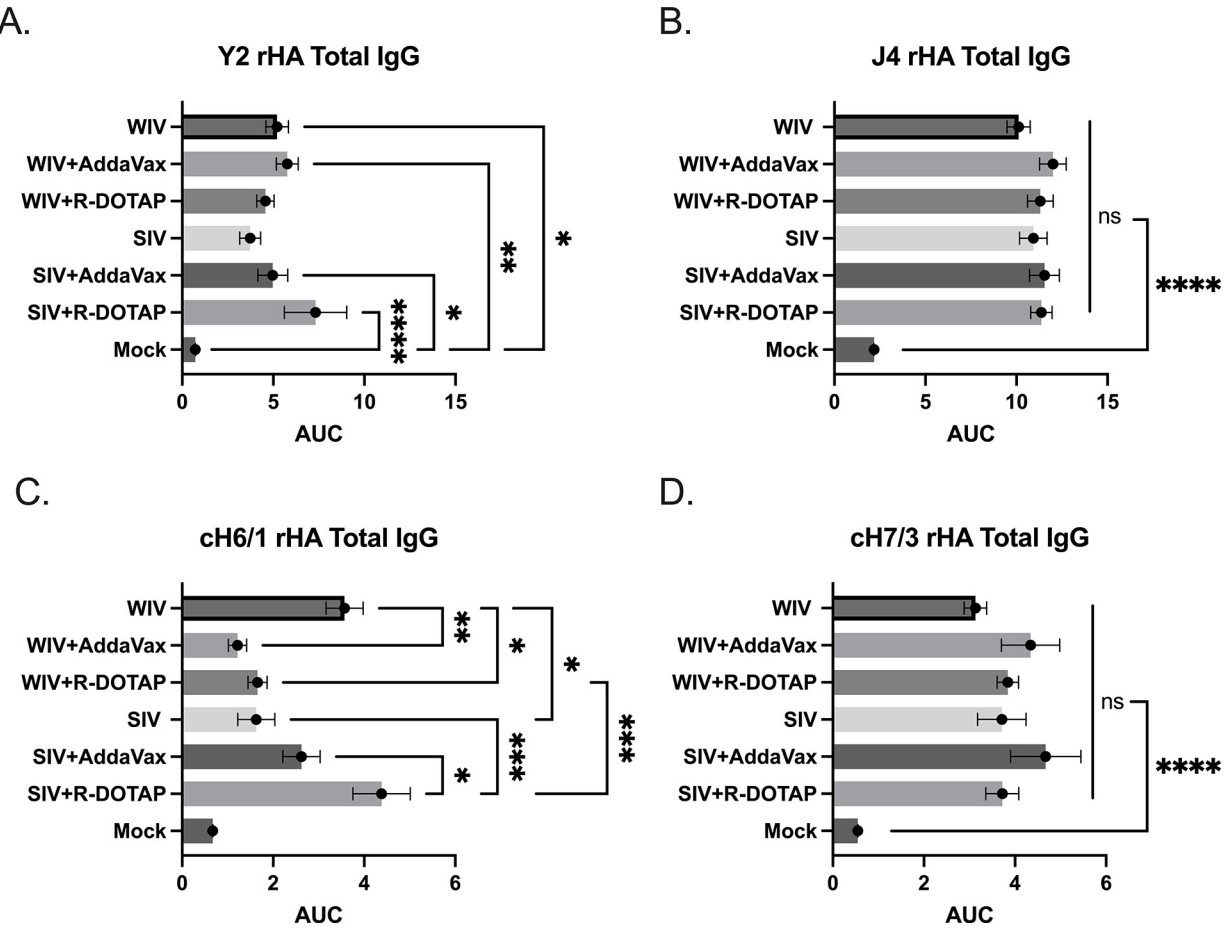

**Fig 4. IgG levels stimulated by COBRA-IIVs 2 weeks post vaccination.** A. Y2-specific total IgG level. B. J4-specific total IgG level. C. H1 HA stem-binding IgG level. D. H3 HA stem-binding IgG level. The X-axis represents different vaccines. The Y-axis represents the Log2 MNA endpoint with absolute mean values ± SEM. The X-axis represents the area under curve (AUC) obtained OD141 values from 3-fold serially diluted sera plus SEM. The Y-axis represents different vaccines. cH6/1: chimeric rHA with H6 head from H6 A/Mallard/Sweden/81/2002 and stalk from A/California/07/2009. cH7/3: chimeric rHA with H7 head from A/Anhui/1/2013 and H3 stalk from A/Texas/50/2012. A P value of less than 0.05 was defined as statistically significant (*, P < 0.05; **, P < 0.01; ***, P < 0.001; ****, P < 0.0001).

HA head and H1 HA stem; cH7/3: chimeric HA with H7 HA head and H3 HA stem) were used. The results showed that ferrets vaccinated with the COBRA-SIV vaccine plus R-DOTAP had more antibodies directed to the group 1 HA stem region compared to ferrets vaccinated with the COBRA-SIV vaccine only or with the COBRA-SIV vaccine plus AddaVax (Fig 4C). In contrast, ferrets vaccinated with the COBRA-WIV vaccine only had significantly higher to the group 1 stem HA compared to ferrets vaccinated with these same vaccines with either adjuvant. Additionally, the COBRA-WIV vaccine only stimulated significantly more group 1 stem-binding antibodies than the COBRA-SIV vaccine only. Overall, all ferrets had similar group 2 anti-HA stem antibodies following vaccination with either vaccine with or without adjuvants (Fig 4D). Taken together, one vaccination of either COBRA-IIV vaccine elicits anti-HA head and stem antibodies following vaccination.

## COBRA-IIVs mitigated clinical signs and viral shedding in pre-immune ferrets after CA/09 exposure

At 14 weeks post-vaccination, vaccinated ferrets were challenged via intranasal inoculation with the H1N1 influenza virus, CA/09 ($1*10^6$PFU/ferret). Unvaccinated ferrets rapidly lost weight following the challenge losing ~15% of their original body weight by day 7 post-infection (Fig 5A) with a third of the ferrets succumbing to infection (Fig 5B). All ferrets vaccinated with COBRA-WIV vaccines, with or without adjuvant, lost between 5–10% body weight in the first 3–4 days post-infection and then, began returning to their original weight over the 14 days of observation (Fig 5A). All COBRA-WIV vaccinated ferrets survived challenge (Fig 5B). Similar results were observed in ferrets vaccinated with COBRA-SIV vaccine with or without adjuvant, except a third of the ferrets vaccinated with COBRA-SIV only vaccines died from challenge (Fig 5B). During the 14-day period, mild clinical symptoms were observed in all ferrets, such as sneezing and nasal discharge and several ferrets in the control group were lethargic (Fig 5C).

Naïve ferrets had high viral nasal wash titers ($1*10^6$ to $1*10^7$ PFU/ml) at 1-day post-infection (Fig 6). Ferrets vaccinated with any vaccine or adjuvant had similar viral nasal wash titers at 1-day post-infection. Influenza virus was detected from the nasal washes of ferrets following the CA/09 challenge (Fig 6). Viral nasal wash titers dropped a log by day 3 post-infection in naïve ferrets (Fig 6) and dropped an additional log to $1*10^4$ PFU/ml at day 5 post-infection. In contrast, vaccinated ferrets had viral nasal wash titers that ranged on average from $1*10^{*3}$ to $1*10^4$ PFU/ml on day 3 post-infection, which was 1 log lower than in naive ferrets (Fig 6). By day 5 post-infection, there was no detectable virus in any COBRA-WIV vaccinated group of ferrets, as well as COBRA-SIV plus R-DOTAP vaccinated ferrets (Fig 6). Low to undetectable viral titers were observed in ferrets vaccinated with COBRA-SIV only or COBRA-SIV plus AddaVax.

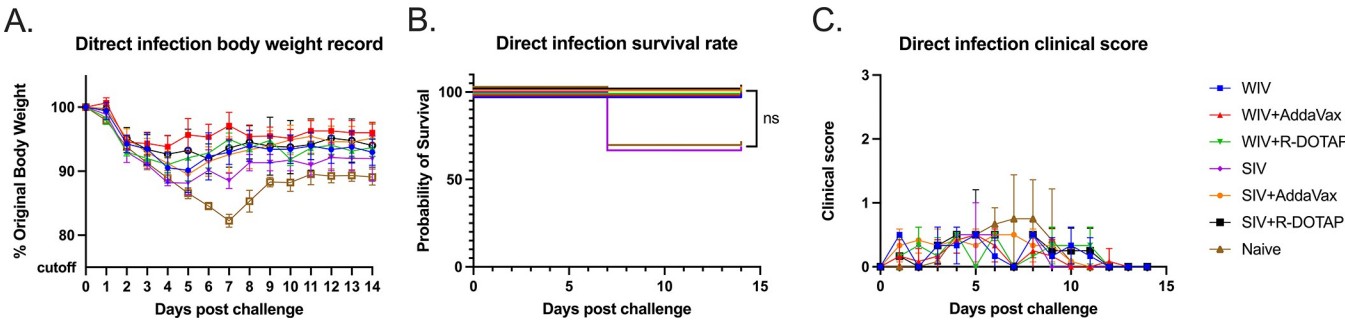

**Fig 5. Development of disease post-direct infection.** A: body weight loss curve. B: survival rate. C. Clinic scores. The legend shows the different vaccine groups with the naïve control with 3 ferrets in each group.

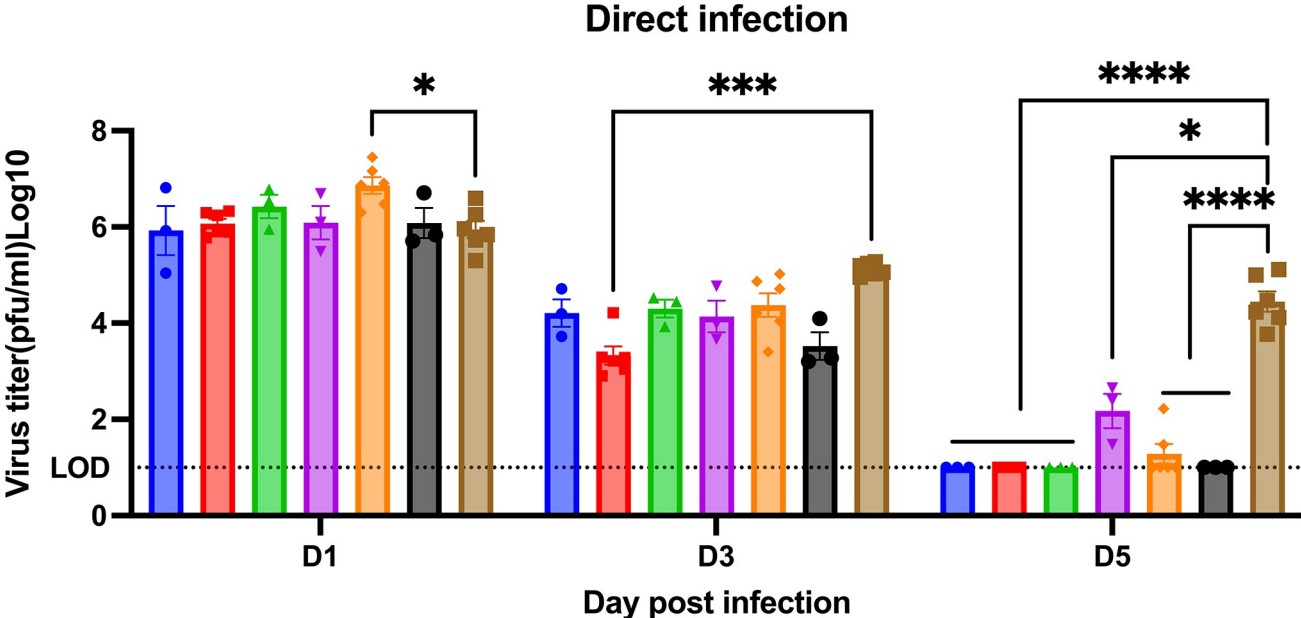

**Fig 6. Viral shedding titer in nasal washes collected post-direct infection.** The X-axis represents the different time points for collecting the nasal washes. The Y-axis represents the viral titer PFU/ml nasal washes in the Log10 scale with absolute mean values ± SEM. The legend shows the different vaccine groups. A P value of less than 0.05 was defined as statistically significant (*, P < 0.05; **, P < 0.01; ***, P < 0.001; ****, P < 0.0001).

To better assess the relationship between the antibody responses and protection, antibody titer, body weight loss, clinical score, and viral shedding titers were recorded for each individual ferret (S1 Table). The results showed that ferrets with higher CA/09-specific antibody titers (HAI>1:40) tended to lose less body weight (<10%) and show fewer clinical symptoms. The viral shedding amount is similar among all ferrets on 1DPI and 3DPI followed by direct infection, but on 5DPI, ferrets with unprotective antibody levels shed more viruses than ferrets with protective antibody levels.

To determine if vaccination with COBRA-IIV vaccines could reduce or block respiratory viral transmission, vaccinated ferrets (transmitters) were challenged and then housed with a new, unvaccinated naïve ferret (receiver) for 14 days (Fig 7A). Receiver ferrets, housed with vaccinated transmitter ferrets, had nasal wash viral titers that ranged on average between $1*10^5$ to $1*10^6$ PFU/ml regardless which vaccinated ferret was co-housed with the naïve receiver ferret (Fig 7B). The naïve receiver ferrets when housed with vaccinated transmitter ferrets lost significant body weight, with ferrets on average losing between 12–20% body weight by day 8–9 post-infection before their weights plateaued or slightly rose by day 14 post-infection (Fig 8A). Two of the 3 receiver ferrets housed with COBRA-WIV only vaccinated transmitter ferrets and 1 of the 3 receiver ferrets housed with COBRA-WIV plus R-DOTAP vaccinated transmitter ferrets succumbed to infection, while all other receiver ferrets survived (Fig 8B). These two groups' clinical scores peaked from 7–10 days post-infection with severe lethargy and sharp body weight loss (>20%), which was even higher than those housed with naïve transmitter ferrets (Fig 8C).

### Contact infection induced less body weight loss and delayed viral shedding than direct infection in COBRA-IIV vaccinated ferrets

To determine if ferret-to-ferret transmission between animals can be reduced or blocked if the receiver ferret was previously vaccinated, new naïve ferrets were infected with CA/09 and then co-housed with a vaccinated ferret (Fig 9). Naïve receiver ferrets had ~$10^{4.5}$ PFU/ml of virus in

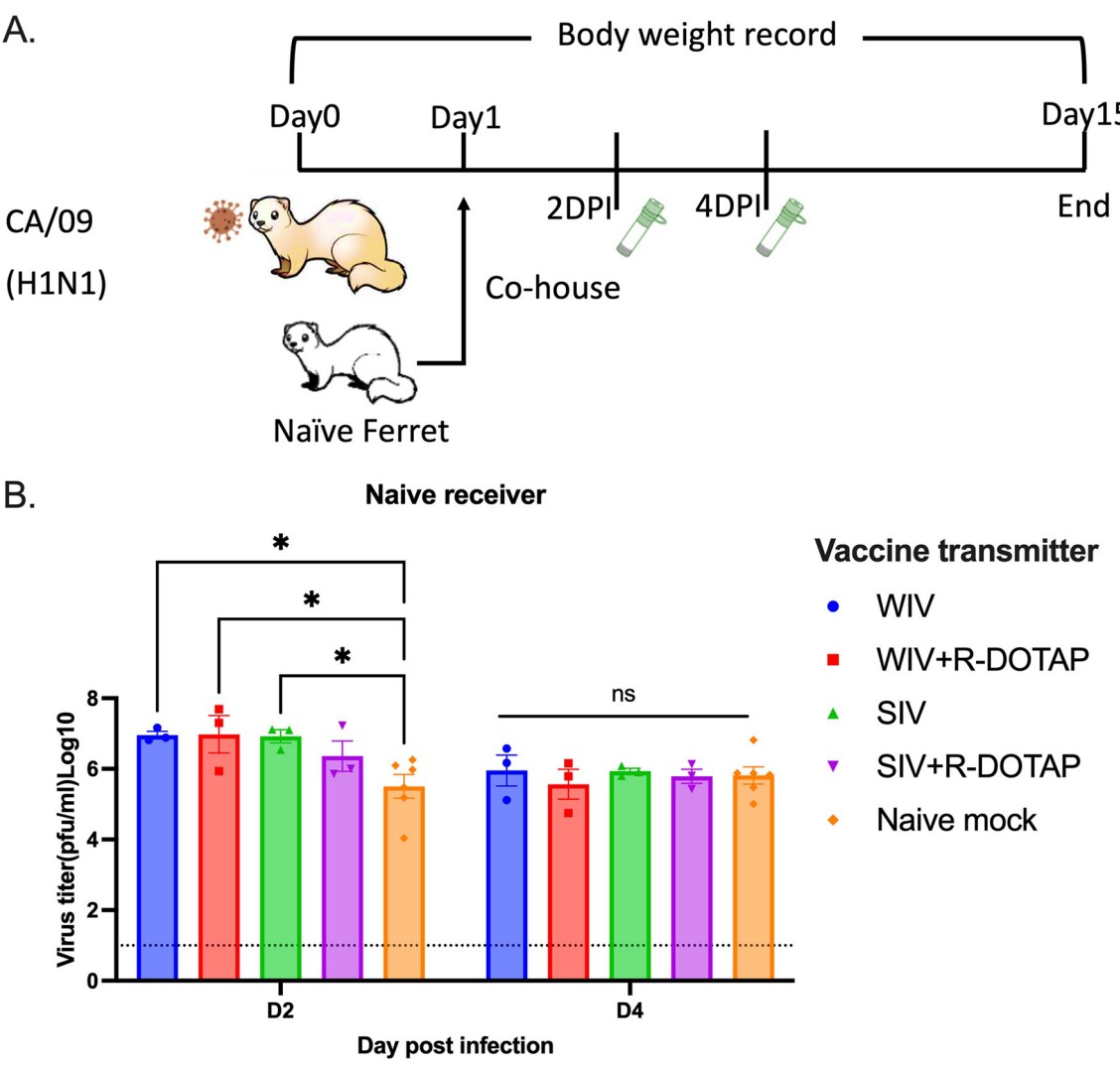

**Fig 7. Transmission study from vaccinated transmitters to naive receivers.** A. experimental design. Vaccinated transmitters were intranasally infected by CA/09 ($1*10^6$PFU/ferret) on day 0. One day later, the naïve receivers were paired with the vaccinated transmitters and they were co-housed for 14 days. The transmission from naïve transmitters to naïve receivers functioned as the control in this study. The nasal washes from the receivers were collected on day 3 and 5, which were considered 2DPI and 4DPI for receivers. Body weight loss, clinical scores, and survival rate were closely monitored for 14 days since co-housing. B. Viral titers in nasal washes collected from receivers at 2DPI and 4DPI. The X-axis represents the different time points for collecting the nasal washes. The Y-axis represents the viral titer PFU/ml nasal washes in the Log10 scale with absolute mean values ± SEM. The legend shows the vaccine that the transmitters got. A P value of less than 0.05 was defined as statistically significant (*, $P < 0.05$; **, $P < 0.01$; ***, $P < 0.001$; ****, $P < 0.0001$).

their nasal washes following 2 days of co-housing with a naïve CA/09 infected transmitter ferret (Fig 9A). These titers rose to ~$10^6$ PFU/ml at day 4 post-infection. Ferrets vaccinated with the COBRA-WIV vaccine only had the lowest viral nasal wash titers (~$10^{2.2}$) at day 2 post-infection vaccination (Fig 9B). All other vaccinated ferrets had slightly higher, but not statistically different, nasal wash viral titers than COBRA-WIV vaccine only vaccinated ferrets. By day 4 post-infection, all ferrets had similar high nasal wash viral titers as unvaccinated ferrets (between $10^5$ and $10^6$ PFU/ml), regardless of the vaccine administered to the receiver ferrets. Ferrets vaccinated with the COBRA-WIV vaccine with or without R-DOTAP lost between 2–5% of original body weight by day 6 post-infection (Fig 9C). In contrast, ferrets vaccinated

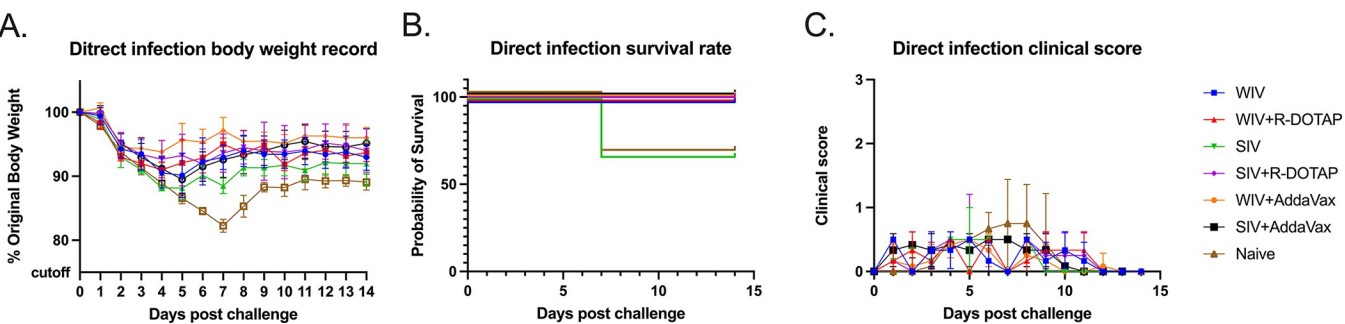

**Fig 8. Development of disease post-direct infection.** A: body weight loss curve. B: survival rate. C. clinical sores. The legend shows the naïve ferrets co-housed with different vaccine groups with the naïve ferrets serving as a control with 3 ferrets in each group.

with COBRA-SIV vaccines had similar weight loss as unvaccinated ferrets losing ~10% of their original body weight by day 6 post-infection. The COBRA-SIV vaccinated ferrets began to recover weight over the next 7 days, whereas the unvaccinated ferrets lost between 13–15% of

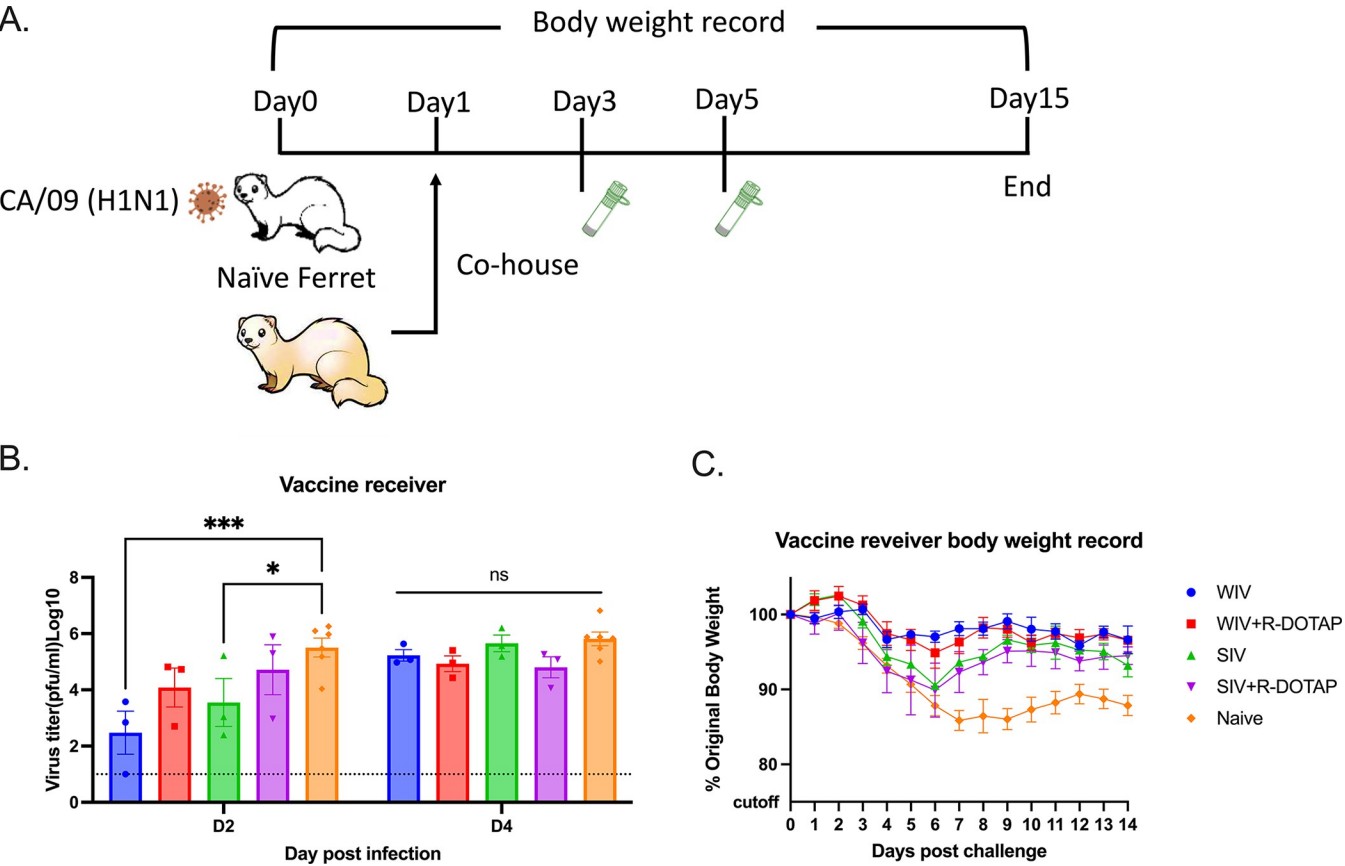

**Fig 9. Transmission study from naïve transmitters to vaccinated receivers.** A. experimental design. Naïve ferrets were intranasally infected by CA/09 ($1*10^6$PFU/ferret) on day 0. One day later, the naïve transmitters were paired with the vaccinated receivers and they were co-housed for 14 days. The transmission from naïve transmitters to naïve receivers functioned as the control in this study. The nasal washes from the receivers were collected on day 3 and 5, which were considered 2DPI and 4DPI for receivers. Body weight loss, clinical scores, and survival rate were closely monitored for 14 days since co-housing. B. Viral titers in nasal washes collected from receivers at 2DPI and 4DPI. The X-axis represents the different time points for collecting the nasal washes. The Y-axis represents the viral titer PFU/ml nasal washes in the Log10 scale with absolute mean values ± SEM. C. body weight loss for receivers post-contact infection. The legend shows the different vaccine groups. A P value of less than 0.05 was defined as statistically significant (*, P < 0.05; **, P < 0.01; ***, P < 0.001; ****, P < 0.0001).

original body weight and plateaued without recovering body weight over the next 7 days of observation (Fig 9C). All vaccinated ferrets only showed mild symptoms after exposure to CA/ 09 (S3 Fig). The individual data shows that even with no detectable CA/09-specific antibodies, some vaccinated ferrets were still protected from the lethal CA/09 challenge and lost <10% of their original body weight (S1 Table), which underscores the activation of cellular immunity in protection.

Compared to directly infected ferrets, the average body weight loss was less in contact-infected naïve ferrets or ferrets vaccinated with WIV, WIV plus R-DOTAP, and SIV, but similar for ferrets vaccinated with SIV plus R-DOTAP (S4 Fig). There was a gap in body weight loss clearly shown between direct infection and contact infection for WIV plus R-DOTAP and WIV vaccinated ferrets, even though no statistical difference was shown (S4 Fig). Besides, the naïve ferrets only showed mild clinical signs, such as sneezing and nasal discharge, after the contact infection, but showed some lethargy after the direct infection.

## Discussion

In 2018, the National Institute of Allergy and Infectious Disease (NIAID) launched a strategy to develop a universal influenza vaccine to ameliorate the global influenza burden [25]. To meet this urgent goal, the COBRA methodology used to generate HA and NA was applied to various vaccine formats for preclinical studies to test their effectiveness [14, 26, 27]. In this study, pre-immune ferrets and transmission models were used to determine the vaccine effectiveness of inactivated influenza virus vaccines expressing COBRA HA proteins. Promisingly, the COBRA-IIV vaccines elicited broadly neutralizing protective antibodies against multiple H1N1 and H3N2 influenza virus strains following a single vaccination in pre-immune ferrets.

In previous mouse studies, COBRA-IIV vaccines were effective in both immunologically naïve and pre-immune animals [20]. In this study, ferrets were used to further assess these vaccine candidates because the mouse model has some limitations. Typically, mice are resistant to infection by most human influenza strains [28], thereby, mouse-adapted strains are often required for the challenge of vaccinated mice [29]. Moreover, mice do not show similar clinical symptoms as humans or poorly transmit viruses between mice [30]. Ferrets, another primary animal model for influenza vaccine research, are naturally susceptible to influenza viruses and develop similar clinical signs as humans, such as sneezing, nasal discharge, fever, and lethargy [31]. More importantly, the human influenza strains transmit efficiently among ferrets, which makes ferrets ideal for studying viral transmission [32–34].

The WIV was first used as the influenza vaccine in the 1940s and showed effective protection against influenza virus infection in humans [16]. The WIV vaccine maintains conformational structures of viral surface glycoproteins and internal gene products [35]. However, the strong immunogenicity was associated with inflammation at the site of administration resulted in limited human use in the U.S. [36]. The SIV vaccines with improved safety have been commonly used as a seasonal influenza vaccine for decades and is safe and effective in most populations [17]. However, the SIV is not as effective in children and the elderly [21, 37, 38]. The reduced vaccine effectiveness elicited by SIV vaccines compared to WIV vaccines was also observed in this study. The COBRA-WIV elicited antibodies with broad HAI activity (9 out of 11 tested strains), whereas the COBRA-SIV elicited HAI activity at >1:40 against 3 out of 11 strains in the HAI panel (Fig 2). Even though the COBRA-SIV stimulated some neutralizing antibodies in several ferrets, overall, it was insufficient to elicit neutralizing antibodies with only one vaccination (Fig 3). Meanwhile the COBRA-WIVs and adjuvanted COBRA-SIVs all elicited significantly higher neutralizing antibody levels than the mock control, which emphasizes the significance of adjuvant for the SIV format. Unexpectedly, the mock control ferrets

showed neutralization activities (Fig 3), while none of them showed HAI activities against the same strains. A similar result that the Pan/99 could elicit HK/19-specific neutralizing antibodies in the mock-vaccinated ferrets [39]. A possible explanation is the NA-specific antibody elicited by the pre-infection of Pan/99, which caused cross-neutralization. This study also compared the antibody levels in each ferret, and found that the undetectable HAI titer was not necessarily associated with low neutralizing antibody level (S1 Table). This evidence indicates that the COBRA-IIVs were able to stimulate not only the antibodies targeting the HA head domain but also other potential epitopes. The HA stem is a potential target and the HA stem-binding antibodies were proven to neutralize live viruses [40].

To test this hypothesis, this study later measured the HA stem-binding antibodies in the serum samples, and the results showed the COBRA-WIV and COBRA-SIV plus R-DOTAP stimulated significantly more HA stem-binding antibodies than the mock control (Fig 4). With the optimized inactivation bioprocess and purification methods, the adverse effects of WIV vaccines can be diminished [18]. In this study, no vaccination-associated reactions were observed after administrating the COBRA-WIV vaccines to ferrets. To enhance the vaccine effectiveness of SIV vaccines, a higher dose of vaccine or addition of an adjuvant has been used in commercial split-inactivated influenza vaccines for the elderly [41]. Similarly, when the COBRA-SIV vaccines were mixed with R-DOTAP adjuvant, the HAI breadth (Fig 2), neutralizing antibody levels (Fig 3) and protection against viral infection (Fig 5) were improved. Meanwhile, the AddaVax did not significantly enhance the vaccine effectiveness compared to unadjuvanted vaccine. Therefore, the utilization of COBRA-SIV may require an adjuvant if tested in the clinic, and the COBRA-WIV showed superiorities for clinic usage.

R-DOTAP is designed to attach to the negative-charged cell membrane and deliver the peptide or protein antigens into the host cells, which stimulates cellular immunity and cross-presentation [42]. R-DOTAP adjuvanted COBRA rHA vaccines elicit antibodies with wider HAI breadth and stimulated more multifunctional T cells (CD8+ and CD4+ T cells) than unadjuvanted COBRA rHA vaccines [24]. The addition of R-DOTAP to COBRA-SIV improved the antibody activity spectrum (Fig 2F) with only one vaccination and induced more stem-binding antibodies than unadjuvanted COBRA-SIV or AddaVax adjuvanted COBRA-SIV (Fig 4B). Compared to AddaVax, R-DOTAP is likely a better adjuvant for COBRA-SIV formulation, since it further improved the breadth HAI activity elicited by COBRA-SIV (Fig 2E and 2F), and was more effective at suppressing viral replication following infection in ferrets (Fig 5). Furthermore, the R-DOTAP was reported to stimulate more durable CD4+ and CD8+ T cells with poly-function than AddaVax, and, these T cell receptors (TCRs) also recognized more distinct influenza peptides [24]. Future studies will evaluate the cellular immunity elicited by COBRA-IIV vaccines and the map TCR usage in order to optimizing the vaccine formulation.

However, either adjuvant showed few positive or even negative effects on the COBRA-WIV in eliciting stem-binding antibodies (Fig 4B), which could result from the self-adjuvant effects of WIV [43, 44]. Since the COBRA-WIV stimulated robust immune responses by itself, the addition of an adjuvant did not introduce significant improvement. A similar result was also noticed in the mouse study that the COBRA-WIV and COBRA-WIV plus AddaVax elicited comparable HAI titers and breadth in naïve mice after two vaccinations [20]. However, the addition of an adjuvant sustained the antibody circulation for both COBRA-WIV and COBRA-SIV (S2 Fig). At 14 weeks post-vaccination, the COBRA-WIV plus AddaVax and COBRA-SIV plus R-DOTAP maintained a protective HAI level against CA/09, while other formulations all waned below 1:40 (S2 Fig). Additionally, in terms of body weight loss and clinical symptoms, the addition of an adjuvant provided better protection against the lethal challenge either by direct infection or contact infection (Figs 6 and 9C and S3 Fig).

To mimic person-to-person transmission patterns in humans, this study employed the ferret co-housing model that allowed contact infection from infected ferrets to uninfected ferrets. Sixty-six percent of immunologically naïve ferrets survived challenge when directly infected with CA/09 (H1N1) (Fig 6B). These ferret lost more weight post-infection compared to ferrets infected by contact transmission. In addition, 100% of ferret survived infection via contact transmission from another ferret. Interestingly, the different infection methods also led to different outcomes of the same vaccine (S4 Fig). The COBRA-SIV vaccinated ferrets had higher fatality and more body weight loss than the COBRA-SIV plus R-DOTAP following direct infection (Fig 6A and 6B). However, after contact infection, these two vaccines elicited similar protective immunity that resulted in improved survival and less weight loss (Fig 9C). This may be a result that the receiver ferrets were exposed to the gradually increasing doses virus during transmission. In contrast, the directly infected ferrets were inoculated with a lethal dose of the virus, which causes acute infection [45]. The infection methods led to different interpretations for the vaccine effectiveness of COBRA-SIV and COBRA-SIV plus R-DOTAP, which emphasizes the importance of model choice in preclinical vaccine studies.

The goal of vaccination is not only to protect the vaccinated individuals against disease and illness but also to reduce shedding and transmission to non-vaccinated people. This is known as herd immunity. The COBRA-IIVs provided protection for over 14 weeks, however, vaccinated ferrets consistently shed virus that was similar to viral titers as naïve ferrets at 1-day post-infection (Fig 6). This was also observed in the previous ferret and pig studies [33, 46]. One possible explanation is that the COBRA-IIV vaccines did not elicit sufficient antibodies to neutralize all viruses. As shown in S2 Fig, the CA/09-specific antibodies in most ferrets dropped below the protective level. Therefore, those viruses still infected the vaccinated ferrets and replicated rapidly in their respiratory tract. However, significantly faster elimination of invaded viruses was observed in the vaccinated ferrets than in the naïve control, which indicates a reduced viral shedding amount in total. The CA/09 is a highly transmissible virus via aerosols or droplets containing only a low viral load [47] and the naïve receivers were exposed to the transmitters for a long time, which leads to infection in the naïve receivers (Fig 8B).

Even though COBRA-IIV vaccines did not block transmission in this study, the period of viral shedding in vaccinated ferrets was shorter than in naïve ferrets (Fig 6). All vaccinated ferrets did not shed virus or significantly less viruses than the naïve control at 5 days post-infection (Fig 6), even for ferrets that had HAI antibody levels >1:40 prior to challenge (S2 Fig). This may indicate that the immune memory was activated and responded rapidly to eliminate those invaded viruses within 5 days. Furthermore, vaccinated ferrets infected through contact had delayed shedding patterns and one ferret (T1987) did not shed any virus at 2 days post-infection (Fig 9B). Both the timing of the exposure and the period of exposure are important factors in determining transmission. The co-housing transmission model does not effectively mimic the public transmission pattern of human model with potentially longer exposure times via droplet, aerosol, and direct contact. However, the primary source of influenza virus transmission in the human population is via droplets or aerosol containing infectious particles with a short exposure time [48]. With shorter exposure time, the viral shedding in COBRA IIV-vaccinated ferrets may be blocked. For example, if a naïve ferret is exposed to T1987 at 1–2 days post-infection for a short time and then separated, the transmission will possibly be blocked because it shed no virus at 2 days post-infection (Fig 7B). Moreover, direct contact introduces more infectious particles than droplets or aerosols [49]. Therefore, in future studies, a more sophisticated transmission model in which the receivers are exposed to the infected transmitters without direct contact in a short time period should be explored. In this way, the effectiveness of COBRA-IIV vaccines to block viral transmission from vaccinated transmitters to naïve receivers can be further assessed. Additionally, designing different timing(s) of exposure,

especially before the clinical signs appear will provide more details about the contagious dynamics after influenza virus infection (https://www.cdc.gov/flu/about/keyfacts.htm).

In humans, HAI titers induced by vaccination that are >1:40 are associated with 50% protection against influenza virus infection [50]. In this study, ferrets that received COBRA-WIV and COBRA-SIV plus R-DOTAP maintained a protective HAI titer against CA/09 before challenge, but these ferrets were not protected from the CA/09 challenge, which may associated with the dose of virus dose used in this study. The different exposure doses of influenza virus lead to different immune responses and consequentially effect vaccine effectiveness [51]. Influenza virus dose as low as $1.95*10^3$ viral copies can effectively infect people [52], which is 3 logs lower than the dose ($10^6$ infectious particles/ferret) used to infect ferrets in this study. This high challenge dose is to ensure the experimental infection and transmission, which may limit the vaccine induced protective immune responses elicited by COBRA-IIV vaccines. Future studies can address various doses of vaccine and adjuvants, as well as various challenge doses of virus.

In summary, COBRA IIV vaccines elicited broadly reactive antibodies and provided long-term protection with only one dose in pre-immune ferrets. Among all tested formulations, the COBRA-WIV showed the most promising effectiveness, as well as its ease of manufacturing (no extra splitting process) and no need for an adjuvant. Additionally, the COBRA-SIV plus R-DOTAP provided comparable protection against the lethal challenge as the COBRA-WIV. Considering that the current influenza virus vaccines are manufactured in the SIV format, the COBRA-SIV plus R-DOTAP is also a promising candidate for humans. Overall, the COBRA-WIV and COBRA-SIV plus R-DOTAP provided broad and durable protection and reduced viral transmission.

## Supporting information

**S1 Fig. Seroconversion post pre-infection.** Sera samples collected 2 weeks post-pre-infection were collected and tested against the historical strains (Sing/86 and Pan/99). The X-axis represents the different strains. The Y-axis represents the Log2 HAI titers with absolute mean values ± SEM. The lower dotted line indicates the HAI titer of 1:40, and the upper dotted line indicates 1:80.
(TIFF)

**S2 Fig. Dynamic HAI level of challenge strain-specific antibody post-vaccination.** Sera samples collected 2, 10, and 14 weeks post-vaccination were tested against the CA/09. The X-axis represents the different sampling time points. The Y-axis represents the Log2 HAI titers with absolute mean values ± SEM. The legend shows the different vaccine groups. The lower dotted line indicates the HAI titer of 1:40, and the upper dotted line indicates 1:80. A P value of less than 0.05 was defined as statistically significant (*, $P < 0.05$; **, $P < 0.01$; ***, $P < 0.001$; ****, $P < 0.0001$).
(TIFF)

**S3 Fig. Vaccinated receivers' clinical scores post-contact infection.** The legend shows the vaccine groups.
(TIFF)

**S4 Fig. Comparison of the body weight loss between post-direct infection and post-contact infection.** A: WIV-vaccinated ferrets. B. WIV+R-DOTAP-vaccinated ferrets. C: SIV-vaccinated ferrets. D. SIV+R-DOTAP-vaccinated ferrets. E. Naïve control ferrets. The blue area represents the body weight loss range post-direct infection. The red area represents the body

weight loss range post-contact infection.
(TIFF)

**S1 Table.**
(DOCX)

## Acknowledgments

We would like to thank James Allen, Ivette Nuñez, and Ying Huang for designing COBRA HA vaccines and Spencer Pierce for purifying the HA antigens. We thank Naoko Uno technical assistance. We also thank the University of Georgia Animal Resource staff, technicians, and veterinarians for their excellent animal care. We also appreciate Benjamin Chadwick for proof-reading and language editing for this paper.

## Author Contributions

**Conceptualization:** Ted M. Ross.

**Data curation:** Hua Shi, Xiaojian Zhang.

**Formal analysis:** Hua Shi.

**Funding acquisition:** Ted M. Ross.

**Investigation:** Hua Shi, Xiaojian Zhang.

**Methodology:** Hua Shi.

**Project administration:** Hua Shi, Ted M. Ross.

**Supervision:** Ted M. Ross.

**Writing – original draft:** Hua Shi.

**Writing – review & editing:** Hua Shi, Xiaojian Zhang, Ted M. Ross.

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
