## [Decision Letter · Decision Letter 0]

13 Sep 2024

PONE-D-24-30084A Single Dose of Inactivated Influenza Virus Vaccine Expressing COBRA Hemagglutinin Elicits Broadly-Reactive and Long-Lasting ProtectionPLOS ONE

Dear Dr. Ross,

Thank you for submitting your manuscript to PLOS ONE. After careful consideration, we feel that it has merit but does not fully meet PLOS ONE’s publication criteria as it currently stands. Therefore, we invite you to submit a revised version of the manuscript that addresses the points raised during the review process.

During the revision process, please address the comments related to presentation of data, with attention paid to description of groups included and interactions between antibodies and antigens described.

We look forward to receiving your revised manuscript.

Kind regards,

Victor C Huber

Academic Editor

PLOS ONE

Journal Requirements:

“This project has been funded as part of the Collaborative Influenza Vaccine Innovations Centers (CIVICs) by the National Institute of Allergy and Infectious Diseases, a component of the NIH, Department of Health and Human Services, under contract 75N93019C00052.  TMR is also supported in part as a Georgia Eminent Scholar by the Georgia Research Alliance, GRA-001.”

“We would like to thank James Allen, Ivette Nuñez, and Ying Huang for designing COBRA HA vaccines and Spencer Pierce for purifying the HA antigens. We thank Naoko Uno technical assistance. We also thank the University of Georgia Animal Resource staff, technicians, and veterinarians for their excellent animal care. We also appreciate Benjamin Chadwick for proofreading and language editing for this paper. This project has been funded as part of the Collaborative Influenza Vaccine Innovations Centers (CIVICs) by the National Institute of Allergy and Infectious Diseases, a component of the NIH, Department of Health and Human Services, under contract 75N93019C00052.  TMR is also supported in part as a Georgia Eminent Scholar by the Georgia Research Alliance, GRA-001.”

“This project has been funded as part of the Collaborative Influenza Vaccine Innovations Centers (CIVICs) by the National Institute of Allergy and Infectious Diseases, a component of the NIH, Department of Health and Human Services, under contract 75N93019C00052.  TMR is also supported in part as a Georgia Eminent Scholar by the Georgia Research Alliance, GRA-001.”

5. Please amend the manuscript submission data (via Edit Submission) to include author Dr. Xiaojian Zhang.

Reviewers' comments:

Reviewer's Responses to Questions

**Comments to the Author**

1. Is the manuscript technically sound, and do the data support the conclusions?

Reviewer #1: No

Reviewer #2: Yes

2. Has the statistical analysis been performed appropriately and rigorously? 

Reviewer #1: I Don't Know

Reviewer #2: Yes

3. Have the authors made all data underlying the findings in their manuscript fully available?

Reviewer #1: Yes

Reviewer #2: Yes

4. Is the manuscript presented in an intelligible fashion and written in standard English?

Reviewer #1: Yes

Reviewer #2: Yes

5. Review Comments to the Author

Reviewer #1: The authors report on the immunogenicity and protective potential of WIV and SIV preparations, adjuvanted or not, of computationally optimized broadly reactive antigen preparations in H1N1 and H3N2 infection primed ferrets. In general the WIV preparations protect the animals better than SIV preparations against CA/09 challenge (direct or through contact transmission).

Major remarks:

1. Line 131: the authors should state the scale of the MDCK cultures that were used for the COBRA vaccine virus preparations and whether their HA was sequence verified to exclude any cell culture adaptation of the COBRA-based viruses.

2. Line 168 and figure 1: correct that all ferrets were infected with a mixture of Sing86 and Pan99 viruses in week -8? Please clarify. Also, please provide the outcome of the seroconversion analysis against the challenge viruses in week-6 serum samples. The timeline in Figure 1 should be scaled proportionally and the figure should be adapted such that it is clear that naïve as well as mixed (?) infection primed ferrets were vaccinated in week 0.

3. Figure 3: serum from mock vaccinated ferrets shows MN activity against HK/19, SA/19, and TAS/20. Please clarify. Where these ferrets primed by infection in week -8 and subsequently mock-vaccinated? If so, this should be made clear in figure 1. Correct then that there were 2 control groups: infection primed followed by mock (PBS) vaccination and naïve (mock infected) followed by mock (PBS) vaccination?

4. Figures 5-9: control groups are confusing: naïve and naïve-mock were differently treated? Infection primed only ferrets included?

5. Figure 5 and 8 should specify the number of animals in each group.

Other remarks:

1. Line 48: please check the wording: “… influenza vaccine aims to develop a vaccine …” does not sound right.

2. Line 91: “immunogenicity” instead of “antigenicity”?

3. Line 138: please define BPL.

4. Please provide the source of BioBeads (line 150).

5. Line 353: prime vaccination implies that there was a boost vaccination as well. Please adapt.

6. Line 380 and 392: duplicated figure 3 legend, please adapt.

Reviewer #2: The manuscript describes the in vivo studies on COBRA antigens for influenza in ferrets. Overall, the data provide a substantive and well-supported evaluation of the candidate vaccines, and this review found the transmission experiments to add quite a bit of novelty. A few suggestions are provided to improve the ms:

1. It appears the vaccine groups conferred a trend toward protection from mortality and morbidity relative to mock control, but the differences don't appear to be statistically significant in the survival graphs. Please indicate this with "ns" notation near the mock control.

2. Table 2 has a lot of information and I wonder if it can be simplified to the key points relevant to the text, or perhaps even moved to the supplement.

3. It is not clear how the authors were able to specifically determine HA-stem antibodies in Figure 4. How do you know these titers reflect only stem-binding antibodies? Please explain the rationale in the main text.

6. PLOS authors have the option to publish the peer review history of their article (what does this mean?). If published, this will include your full peer review and any attached files.

Reviewer #1: No

Reviewer #2: No

---

## [Author Response · Author response to Decision Letter 0]

23 Sep 2024

September 18, 2024

Editor, PLoS One

Dear Editor: 

Thank you for offering us the opportunity to revise and resubmit the manuscript ‘A Single Dose of Inactivated Influenza Virus Vaccine Expressing COBRA Hemagglutinin Elicits Broadly-Reactive and Long-Lasting Protection’ to the Plos One Journal. We appreciate the Editor’s and Reviewers’ insightful suggestions that were helpful in improving our paper. Tracked Changes and highlighted in the revised manuscript. Also, please see below for a point-by-point response to each comment and concern.

Comment 1. Please ensure that your manuscript meets PLOS ONE's style requirements, including those for file naming. The PLOS ONE style templates can be found at

Response: Thank you for your detailed instructions. The format has been checked and meets the requirements. 

Comment 2. Thank you for stating the following financial disclosure:

“This project has been funded as part of the Collaborative Influenza Vaccine Innovations Centers (CIVICs) by the National Institute of Allergy and Infectious Diseases, a component of the NIH, Department of Health and Human Services, under contract 75N93019C00052. TMR is also supported in part as a Georgia Eminent Scholar by the Georgia Research Alliance, GRA-001.”

Response: Thank you for pointing this out. The financial disclosure has been added with the statement of the funders’ roles at the end of the manuscript. It is highlighted. Please check on page 34. 

Comment 3. Thank you for stating the following in the Acknowledgments Section of your manuscript:

“We would like to thank James Allen, Ivette Nuñez, and Ying Huang for designing COBRA HA vaccines and Spencer Pierce for purifying the HA antigens. We thank Naoko Uno technical assistance. We also thank the University of Georgia Animal Resource staff, technicians, and veterinarians for their excellent animal care. We also appreciate Benjamin Chadwick for proofreading and language editing for this paper. This project has been funded as part of the Collaborative Influenza Vaccine Innovations Centers (CIVICs) by the National Institute of Allergy and Infectious Diseases, a component of the NIH, Department of Health and Human Services, under contract 75N93019C00052. TMR is also supported in part as a Georgia Eminent Scholar by the Georgia Research Alliance, GRA-001.”

“This project has been funded as part of the Collaborative Influenza Vaccine Innovations Centers (CIVICs) by the National Institute of Allergy and Infectious Diseases, a component of the NIH, Department of Health and Human Services, under contract 75N93019C00052. TMR is also supported in part as a Georgia Eminent Scholar by the Georgia Research Alliance, GRA-001.”

Response: Thank you for your guiding. The funding information has been removed from the acknowledgment section. The funding statement “This project has been funded as part of the Collaborative Influenza Vaccine Innovations Centers (CIVICs) by the National Institute of Allergy and Infectious Diseases, a component of the NIH, Department of Health and Human Services, under contract 75N93019C00052. TMR is also supported in part as a Georgia Eminent Scholar by the Georgia Research Alliance, GRA-001” is correct. 

Comment 4. Please provide a complete Data Availability Statement in the submission form, ensuring you include all necessary access information or a reason for why you are unable to make your data freely accessible. If your research concerns only data provided within your submission, please write "All data are in the manuscript and/or supporting information files" as your Data Availability Statement.

Response: Thank you for notifying me about this. The data availability statement has been added at the end of this manuscript in highlight. Please check line 746 on page 34. 

Comment 5. Please amend the manuscript submission data (via Edit Submission) to include author Dr. Xiaojian Zhang.

Response: We apologize for the oversight. Dr. Zhang has been added to the submission.

Comment 6. Please include your full ethics statement in the ‘Methods’ section of your manuscript file. In your statement, please include the full name of the IRB or ethics committee who approved or waived your study, as well as whether or not you obtained informed written or verbal consent. If consent was waived for your study, please include this information in your statement as well.

Response: Thank you for pointing this out. The ethics statement has been added in the ‘methods’ section and highlighted. Please check on page 15. 

Comment 7. Please review your reference list to ensure that it is complete and correct. If you have cited papers that have been retracted, please include the rationale for doing so in the manuscript text, or remove these references and replace them with relevant current references. Any changes to the reference list should be mentioned in the rebuttal letter that accompanies your revised manuscript. If you need to cite a retracted article, indicate the article’s retracted status in the References list and also include a citation and full reference for the retraction notice.

Response: References checked all references include a citation and full reference

Reviewers' comments:

Reviewer's Responses to Questions 

Response: Thank you for your detailed guidance. The reference list has been checked and meets the requirements. 

Comments to the Author

Review Comments to the Author

Reviewer #1: The authors report on the immunogenicity and protective potential of WIV and SIV preparations, adjuvanted or not, of computationally optimized broadly reactive antigen preparations in H1N1 and H3N2 infection primed ferrets. In general the WIV preparations protect the animals better than SIV preparations against CA/09 challenge (direct or through contact transmission).

Major remarks:

1. Line 131: the authors should state the scale of the MDCK cultures that were used for the COBRA vaccine virus preparations and whether their HA was sequence verified to exclude any cell culture adaptation of the COBRA-based viruses.

Response: The MDCK cells were seeded in a T-175 cell culture flask. When cell monolayer reached 70-80% confluency, there were ~1.6*10^7 cells in each flask. Twenty flasks were used for each virus. The clarification of the scale of MDCK was added in the manuscript highlighted. Please check on page 7. 

The HA sequence was not verified after amplification in the MDCK cells because we used the same culture environment as the rescuing process. 

2. Line 168 and figure 1: correct that all ferrets were infected with a mixture of Sing86 and Pan99 viruses in week -8? Please clarify. Also, please provide the outcome of the seroconversion analysis against the challenge viruses in week-6 serum samples. The timeline in Figure 1 should be scaled proportionally and the figure should be adapted such that it is clear that naïve as well as mixed (?) infection primed ferrets were vaccinated in week 0.

Response: The description of the naïve ferrets serving as naïve transmitters or naïve receivers was added and highlighted. Please check on page 10. 

The supplemental data was added to show the seroconversion against the pre-infection strains. Please check Suppl Fig. 1. File, that is highlighted in the manuscript at on page 9, line 176-177. 

 The length between each procedure has been adjusted proportionally. The nasal wash time points are adjusted, but not proportionally because of their high frequency. Figure 1 has been revised.

3. Figure 3: serum from mock vaccinated ferrets shows MN activity against HK/19, SA/19, and TAS/20. Please clarify. Where these ferrets primed by infection in week -8 and subsequently mock-vaccinated? If so, this should be made clear in figure 1. Correct then that there were 2 control groups: infection primed followed by mock (PBS) vaccination and naïve (mock infected) followed by mock (PBS) vaccination?

Response: Thank you for the question. The high neutralizing activities in the mock-vaccinated ferrets were unexpected at first. We repeated the assays and got the same results, which confirmed the reliability of our data. We also noticed similar results from other studies (published or under preparation). This results may be due to the similarity between the historical H3N2 strains and recent circulating strains (HK/19, SA/19, and TAS/20). The explanation has been added in the results section (line 383-384, page 19) and discussion section (page 28). 

 Yes, those ferrets were preimmune and mock-vaccinated. A clearer statement has been added in the method section (line 168-181, page 10) and the Figure 1 legend (line 353-358, page 17). 

 There is one mock-vaccinated group serving as the control. The naïve ferrets (no pre-infection, nor vaccination) were only used in the transmission study as naïve transmitters or naïve receivers. 

4. Figures 5-9: control groups are confusing: naïve and naïve-mock were differently treated? Infection primed only ferrets included?

Response: In figures 5-9, the control group indicates non-vaccinated naïve ferrets. All of the pre-immune mock-vaccinated ferrets are written as mock-vaccinated or mock control group. In the challenge study and transmission study, only the naïve ferrets were included as the control group for their severe clinical symptoms. In figures 5-9, the control group has been indicated as naïve in the revised version for clarity.

5. Figure 5 and 8 should specify the number of animals in each group.

Response: Number of animals in each group has been added in the figure legends for figure 5 and 8. Please check page 22 and 24. 

Other remarks:

1. Line 48: please check the wording: “… influenza vaccine aims to develop a vaccine …” does not sound right.

2. Line 91: “immunogenicity” instead of “antigenicity”?

3. Line 138: please define BPL.

4. Please provide the source of BioBeads (line 150).

5. Line 353: prime vaccination implies that there was a boost vaccination as well. Please adapt.

6. Line 380 and 392: duplicated figure 3 legend, please adapt.

Response: Thank you for pointing them out. Modifications have been made to the manuscript. 

Reviewer #2: The manuscript describes the in vivo studies on COBRA antigens for influenza in ferrets. Overall, the data provide a substantive and well-supported evaluation of the candidate vaccines, and this review found the transmission experiments to add quite a bit of novelty. A few suggestions are provided to improve the ms:

1. It appears the vaccine groups conferred a trend toward protection from mortality and morbidity relative to mock control, but the differences don't appear to be statistically significant in the survival graphs. Please indicate this with "ns" notation near the mock control.

Response: Thank you for your suggestion. The ‘ns’ notation has been added to the figure 5B to show the no statistical difference among tested groups for the survival rate. 

2. Table 2 has a lot of information and I wonder if it can be simplified to the key points relevant to the text, or perhaps even moved to the supplement.

Response: Individual data is provided. The table 2 has been moved to the supplemental materials as S1 Table. Please check the supplemental material documents. 

3. It is not clear how the authors were able to specifically determine HA-stem antibodies in Figure 4. How do you know these titers reflect only stem-binding antibodies? Please explain the rationale in the main text.

Response: Anti-stem antibodies were determined using chimeric HAs with the targeted stem and untargeted head domain to coat the ELISA plate. Therefore, the bound antibodies could only bind to the stem region, by which we confirmed the stem-binding antibody levels. More information has been added to the main text, please check the highlighted part at line 412-414, page 20. 

Best regards,

Ted M. Ross

Director – Global Vaccine Development

Florida Research & Innovation Center

9801 SW Discovery Way

Port Saint Lucie, FL 34987 USA

---

## [Editor Report · Decision Letter 1]

25 Sep 2024

A Single Dose of Inactivated Influenza Virus Vaccine Expressing COBRA Hemagglutinin Elicits Broadly-Reactive and Long-Lasting Protection

PONE-D-24-30084R1

Dear Dr. Ross,

We’re pleased to inform you that your manuscript has been judged scientifically suitable for publication and will be formally accepted for publication once it meets all outstanding technical requirements.

Kind regards,

Victor C Huber

Academic Editor

PLOS ONE
---

## [Editor Report · Acceptance letter]

24 Oct 2024

PONE-D-24-30084R1 

PLOS ONE

Dear Dr. Ross, 

I'm pleased to inform you that your manuscript has been deemed suitable for publication in PLOS ONE. Congratulations! Your manuscript is now being handed over to our production team.

Kind regards, 

on behalf of

Dr. Victor C Huber 

Academic Editor

PLOS ONE